# Distinct cellular immune responses in children en route to type 1 diabetes with different first-appearing autoantibodies

Inna Starskaia [1,2,3,14], Milla Valta [3,4,14], Sami Pietilä [1,2,14], Tomi Suomi [1,2,14], Sirpa Pahkuri [3,4], Ubaid Ullah Kalim [1,2], Omid Rasool [1,2], Emilie Rydgren[1,2], Heikki Hyöty[5], Mikael Knip [6,7], Riitta Veijola [8], Jorma Ilonen[4], Jorma Toppari[2,9,10,11], Johanna Lempainen [4,11,12] ✉, Laura L. Elo [1,2,13] ✉ & Riitta Lahesmaa [1,2,13] ✉

Previous studies have revealed heterogeneity in the progression to clinical type 1 diabetes in children who develop islet-specific antibodies either to insulin (IAA) or glutamic acid decarboxylase (GADA) as the first auto-antibodies. Here, we test the hypothesis that children who later develop clinical disease have different early immune responses, depending on the type of the first autoantibody to appear (GADA-first or IAA-first). We use mass cytometry for deep immune profiling of peripheral blood mononuclear cell samples longitudinally collected from children who later progressed to clinical disease (IAA-first, GADA-first, ≥2 autoantibodies first groups) and matched for age, sex, and HLA controls who did not, as part of the Type 1 Diabetes Prediction and Prevention study. We identify differences in immune cell composition of children who later develop disease depending on the type of autoantibodies that appear first. Notably, we observe an increase in CD161 expression in natural killer cells of children with ≥2 autoantibodies and validate this in an independent cohort. The results highlight the importance of endotype-specific analyses and are likely to contribute to our understanding of pathogenic mechanisms underlying type 1 diabetes development.

Type 1 diabetes is one of the most common chronic autoimmune diseases in children. The pathogenesis of the disease is determined by interactions between genetics and poorly defined environmental factors that result in an aberrant immune response against insulin-producing pancreatic β cells[1–4].

An imbalance or shift in different T regulatory (Treg) cell subsets that control pathogenic T cells may contribute to type 1 diabetes development[5]. In particular, defects in the function or frequency of FOXP3[+] Treg cells have been associated with type 1 diabetes[5]. Furthermore, changes in the frequency of circulating activated T follicular helper and CXCR5[-]PD-1[hi] peripheral T helper cells and CD4[+] Tregs have been observed in at-risk children who are positive for ≥2 islet-specific

autoantibodies and newly diagnosed type 1 diabetes participants[6–8]. However, the children in these studies had already developed auto-antibodies and clinical disease.

To improve prediction and prevention of type 1 diabetes, it is important to identify changes in immune cell subsets at early stages of the autoimmune process. We and others have shown that early changes, even before seroconversion, can be detected in children who later develop type 1 diabetes[9–13]. Autoantibodies targeting insulin (IAA) and glutamic acid decarboxylase (GADA) are the most commonly detected first autoantibodies and are associated with the age of a child at the time of seroconversion. The appearance of IAA shows a sharp peak at 1–2 years of age and decreases thereafter. Seroconversion to GADA

peaks around the age of 4–5 years declining thereafter and remaining rather stable during childhood[14]. Children with IAA or GADA as the first autoantibodies also differ in the progression of islet autoimmunity, and these observations indicate heterogeneity in type 1 diabetes pathogenesis[15].

In the present study, we sought to identify changes in the early immune response associated with type 1 diabetes with regard to the type of autoantibody that appears first. We were particularly interested in the composition of the immune cells before the appearance of autoantibodies in IAA-first and GADA-first children. In addition, we examined a group of children who tested positive for more than one autoantibody in the first available sample (≥2 Aab first group). We used mass cytometry for deep single-cell immune profiling of peripheral blood mononuclear cell (PBMC) samples from children who later progressed to clinical disease. In-depth analysis of immune cell subsets might contribute to our understanding of the pathogenic mechanisms underlying the clinical presentation of type 1 diabetes and provide potential biomarkers for disease prediction.

Here, we show that the composition of PBMC differ between IAA-first, GADA-first, and ≥2 Aab first groups showing significant differences in NK cell, B cells, CD4[+], CD8[+] and γδ TCR T cell proportions. These differences are endotype-specific and not seen when samples from all the three groups were anlaysed together, highlighting the importance of endotype-specific analysis. Significant differences between cases and controls are also detected in the expression of proteins in NK cells, CD4[+] T cells and CD8[+] T cells. Finally, the increased expression of CD161 in NK cells in children with ≥2 Aab compared to their controls is validated in an independent cohort of participants by flow cytometry.

## Results
### Clinical characteristics of study participants
To better understand the type 1 diabetes pathophysiology with respect to the type of the first appearing autoantibody, we established a cohort of 29 genetically susceptible children who later progressed to clinical disease (Fig. 1a, Table 1). The study subjects were all participants in the Finnish Type 1 Diabetes Prediction and Prevention study (DIPP)[16]. The

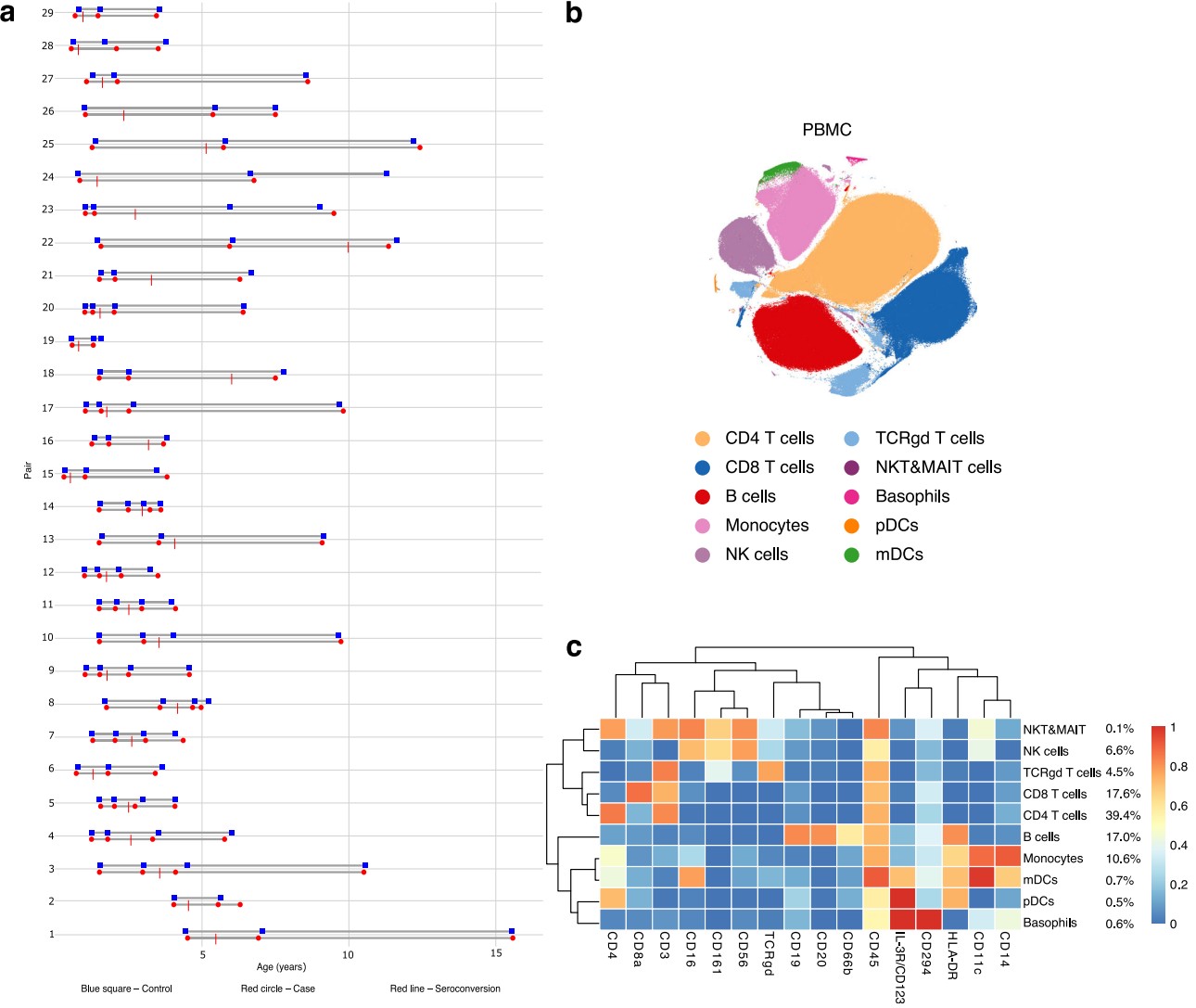

**Fig. 1 | Study design and immune cell composition of the study participants. a** The study cohort. Sample collection from each individual of 29 case-control pairs (y-axis) is visualized on a timeline (age, x-axis). A red circle and a blue square indicate the sample collected with respect to age (x-axis) from a case and a matching control, respectively. Time of seroconversion is indicated with a red crossing line. **b** t-SNE plot of a random subset of 3500 PBMCs from each sample obtained by the mass cytometry analysis and colored to show the main cell types. **c** Heatmap of the normalized marker expression in the main cell types with relative abundances of each cell type. t-SNE t-distributed stochastic neighbor embedding. NK natural killer, NKT&MAIT natural killer T cells and mucosal-associated invariant T cells, mDCs myeloid dendritic cells, pDCs plasmacytoid dendritic cells, PBMC peripheral blood mononuclear cells.

**Table 1 | Clinical characteristics of study participants**

|  | Discovery cohort | | | | Validation cohort | | | |
|---|---|---|---|---|---|---|---|---|
|  | total | IAA | GADA | ≥2 Aab | total | IAA | GADA | ≥2 Aab |
| N, cases | 29 | 11 | 9 | 9 | 35 | 14 | 11 | 10 |
| Sex (F/M) | 8/21 | 1/10 | 3/6 | 4/5 | 20/15 | 6/8 | 9/2 | 5/5 |
| Age at seroconversion (SD) | 2.9 (2.0) | 2.1 (1.2) | 2.8 (1.1) | 4.1 (2.8) | 3.2 (2.7) | 2.7 (1.4) | 4.8 (3.8) | 2.1 (1.8) |
| Age at T1D (SD) | 9.4 (3.3) | 8.6 (2.7) | 10 (3.4) | 9.9 (4.0) | 6.8 (4.6) | 7.6 (4.9) | 8.2 (3.4) | 4.1 (4.5) |
| HLA-DR status (DR3/DR4/both) | 19/22/12 | 7/9/5 | 6/6/3 | 6/7/4 | 19/31/15 | 5/12/3 | 5/9/3 | 9/10/9 |

*N* number of participants, *F* female, *M* male, *SD* standard deviation, *T1D* type 1 diabetes.

mean ages for seroconversion and progression to type 1 diabetes in case subjects in the entire discovery cohort were 2.9 (±2.0) and 9.4 (±3.3) years, respectively. Eleven subjects had IAA as their first appearing autoantibody, nine had GADA, and nine had ≥2 Aab first at time of seroconversion. The respective seroconversion ages in these subgroups were 2.1 (±1.2), 2.8 (±1.1) and 4.1 (±2.8), whereas progression to disease occurred at the ages of 8.6 (±2.7), 10.0 (±3.4) and 9.9 (±4.0) years, respectively. A genetically susceptible but an autoantibody-negative control child matched for age, HLA and sex was selected for each case subject from the DIPP study.

To confirm the observations from the discovery cohort, a separate validation cohort of 30 children (ten cases in each subgroup) who had developed β cell autoimmunity or progressed to clinical type 1 diabetes, was formed (Table 1). These children and their autoantibody-negative controls were also participants in the DIPP study and were selected using the same criteria as in the discovery cohort. The mean ages for seroconversion and type 1 diabetes progression in this cohort were 3.2 (±2.7) and 6.8 (±4.6), respectively. In the IAA-first, GADA-first and ≥2 Aab first groups, the mean ages for seroconversion were 2.7 (±1.4), 4.8 (±3.8) and 2.1 (±1.8), respectively. The respective mean ages for progression to clinical disease in these groups were 7.6 (±4.9), 8.2 (±3.4) and 4.1 (±4.5).

## PBMC composition of study participants

To identify the main PBMC cell subsets and their proportions and to further characterize their properties, we applied a 40-marker mass cytometry panel (Supplementary Table 1). Using unsupervised clustering and t-SNE visualization, we identified ten cell subsets in the PBMC samples (Fig. 1b, c). Further, the cell annotation was consistent with the automated analysis performed with Maxpar Pathsetter™ software (Standard BioTools, USA). Linear mixed effects modeling was applied to study immune cell-type composition changes between type 1 diabetes children and their controls. We observed age-associated changes in most of the identified cell types (Supplementary Fig. 1). For example, the proportions of CD4$^+$ T cells (false discovery rate, FDR = 0.0006) and B cells (FDR = 0.00002) decreased, whereas the numbers of γδ T (FDR = 0.04), monocytes (FDR = 2.7e-14), and other cell types increased with age (Supplementary Fig. 1, Supplementary Table 2), as expected[17,18].

## Immune cell composition differs between the groups

Next, to explore if changes in cell type proportions are specific to an autoantibody group, we performed a direct comparison between the ≥2 Aab first, IAA-first, GADA-first, and the control groups. We discovered differences in cell-type proportions in the ≥2 Aab first and GADA-first groups at FDR ≤ 0.05, whereas no significant differences were found in the IAA-first group (Fig. 2, Supplementary Table 3). We found the proportion of NK cells was significantly higher in the ≥2 Aab first and GADA-first group children than in the controls (FDR < 0.05, Fig. 2a), whereas the opposite was observed for CD4$^+$ T cells in the ≥2 Aab first group (FDR = 0.005, Fig. 2b). Similarly, a decreased proportion of CD8$^+$ T cells was observed in the ≥ 2 Aab first group associated with the disease progression (FDR = 0.0005, Fig. 2c), whereas the

GADA-first group had a higher proportion of CD8$^+$ T cells (FDR = 0.007, Fig. 2c). Additionally, we observed a significant increase in the proportion of γδ T cells in the ≥2 Aab first group (FDR = 0.005, Fig. 2d) and a decreased proportion of B cells in GADA-first group (FDR = 0.046, Fig. 2e).

We also observed significant differences in NKT&MAIT cells and mDCs in the GADA-first group (Supplementary Table 3). However, we did not get sufficient numbers of these cells as their abundance in young children was very low. Further studies would be needed to draw conclusions on these cell types.

Importantly, when we combined all the samples from the three autoantibody groups (all samples group) and analyzed them together, we found no differences in the proportions of these cell types between the children developing type 1 diabetes and controls at FDR ≤ 0.05 (Supplementary Table 2), with the exception of a decreased proportion of CD8$^+$ T cells in cases compared to the controls associated with disease progression (FDR = 0.008, Fig. 2c). The goodness of fit of the models with subset groups was also significantly better than the all samples group models for all major cell types (*p* < 0.05, Supplementary Table 4).

Accordingly, the classification of individuals by the type of autoantibodies that appear first allowed us to observe differences in immune cell compositions which would have been masked without such grouping. Our results are consistent with the idea that the type of autoantibodies that are detected first are associated with a given path to disease.

## CD161 upregulated in NK and CD4$^+$ T cells in ≥2 Aab first group children

We next focused on the expression of markers reflecting a given function in cell subsets. Along with the increased proportion of NK cells, we found CD161 expression in NK cells to be significantly higher in ≥2 Aab first group compared to the controls (FDR = 0.034, Fig. 3a, Supplementary Data 1). Further analysis showed that the proportion of CD161$^+$ cells was elevated in the ≥2 Aab first group children (Fig. 3b). Notably, the difference was prominent at the very early stage of disease development, before seroconversion (Fig. 3c, timepoints 1 and 2). The CD161 levels in the NK cells were not different between the cases and controls of the IAA-, GADA-first groups (Fig. 3a, Supplementary Data 1). In the analysis combining all three autoantibody groups, we did not find any difference in CD161 expression in NK cells (Fig. 3a, Supplementary Data 1). Next, we found the expression levels of CD27, a marker associated with immature phenotype of NK cells, were lower cases of ≥2 Aab first group than in controls (Supplementary Data 1).

Further, CD161 was also upregulated in CD4$^+$CCR4$^+$ (FDR = 0.00006, Fig. 4a, b, Supplementary Data 2) and CD45RA$^+$CCR7$^+$ subsets (FDR = 0.007, Supplementary Data 2) in the ≥2 Aab first group children when compared to controls. The CD45RA$^+$CCR7$^+$ subset constituted up to 84.7% of all CD4$^+$ T cells and had a naive phenotype (Fig. 4a). The proportion of CD161$^+$ ranged from 0.5–2.0% of total CD45RA$^+$CCR7$^+$. The CD161 levels in the same subsets of IAA- and GADA-first groups, and across all samples were not different in cases compared to controls (Fig. 4b, Supplementary Data 2).

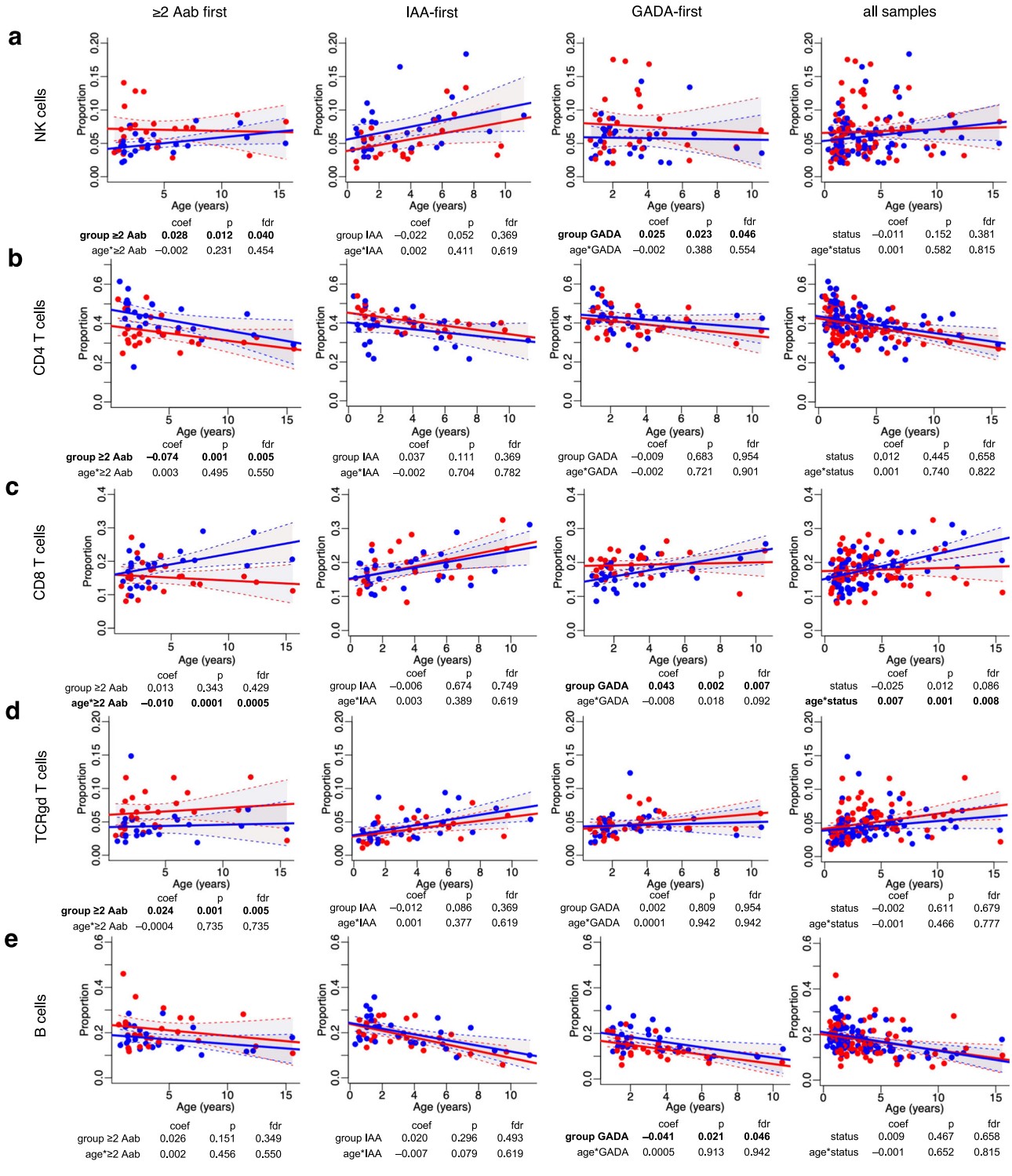

**Fig. 2 | Cell type proportions in the children from the ≥2 Aab first, IAA-first, GADA-first and all samples groups. a** Scatter plots of NK cells proportions over time. **b** Scatter plots of CD4⁺ T cells over time. **c** Scatter plots of CD8⁺ T cells over time. **d** Scatter plots of γδ T cells over time. **e** Scatter plots of B cells over time. Each plot is annotated with the mixed effects model coefficient, *p* value and FDR for the comparison between the ≥2 Aab first, IAA-first, GADA-first, and the control groups (*group*) and their interaction with age (*age\*group*). Bold indicates statistically significant results. Red and blue dots indicate case and control samples, respectively. Solid red and blue lines show linear regression fit and shaded areas show 95% confidence intervals for the predicted values for cases and controls, respectively. Statistical analyses were performed using linear mixed effects modeling. Reported p values were calculated from two-tailed t-tests and multiple testing correction was done using the Benjamini–Hochberg method. NK natural killer, *coef* coefficient, *p p* value, *fdr* false discovery rate.

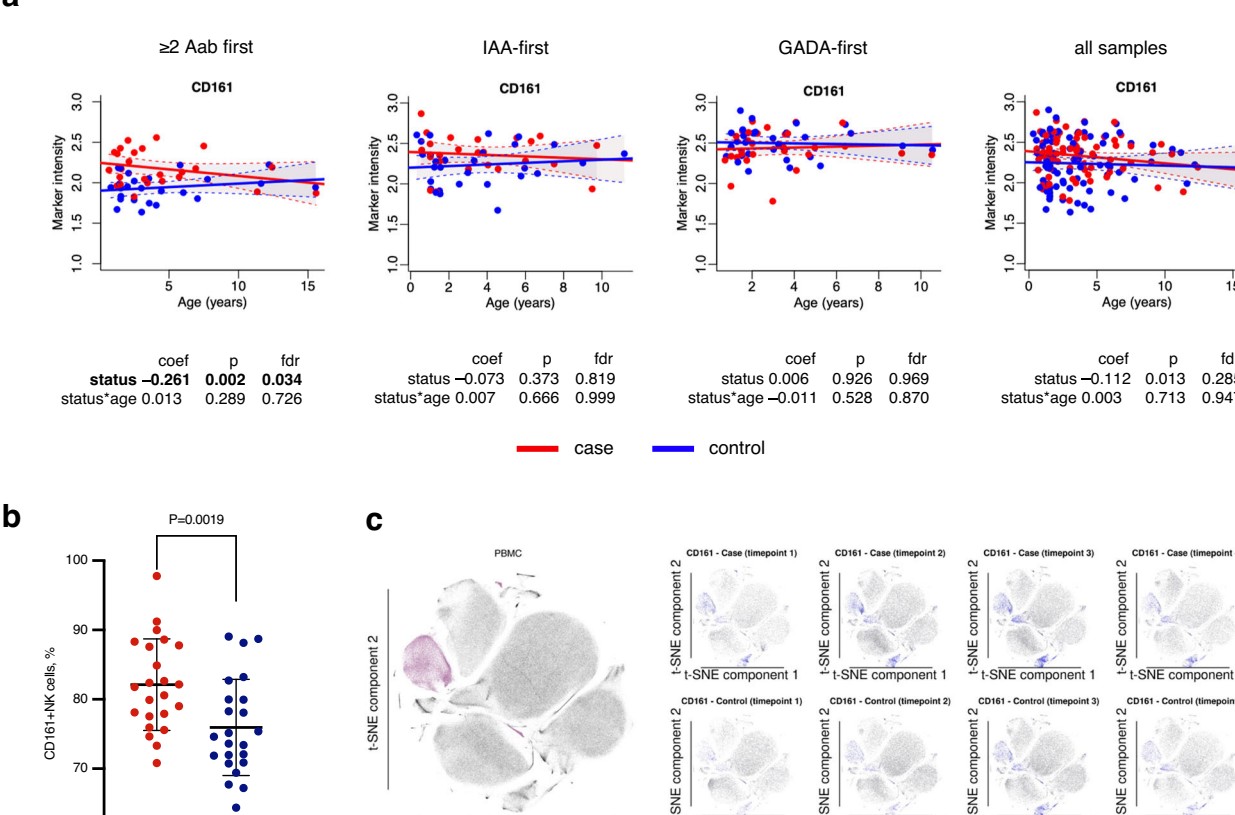

**Fig. 3 | Increased expression of CD161 in NK cells. a** Scatter plot of CD161 expression levels (y-axis) in NK cells over time (x-axis) in the children of the ≥2 Aab first, IAA-first, GADA-first and all samples groups. Each plot is annotated with the linear mixed effects model coefficient, p-value and FDR for the comparison between cases and controls (*status*) and their interaction with age (*status*age*). Red and blue dots indicate case and control samples, respectively. Solid red and blue lines show linear regression fit and shaded areas show 95% confidence intervals for the predicted values for cases and controls, respectively. Statistical analyses were performed using linear mixed effects modeling. Reported p values were calculated from two-tailed t-tests and multiple testing correction was done using the Benjamini–Hochberg method. **b** Strip chart of CD161-expressing NK cell frequencies in cases (n = 23 longitudinally collected samples from 9 individual donors, red) and controls (n = 23 longitudinally collected samples from 9 individual donors, blue) in the ≥2 Aab first group of the discovery cohort. Mean and standard deviation shown on the charts. Statistical analysis was performed using two-tailed paired *t*-test, P value = 0.0019. **c** t-SNE plots showing the normalized CD161 expression (blue color) in PBMC cells of cases (upper plots) and controls (lower plots) at each time point. NK cells are visualized by the violet color on the left t-SNE plot. For each t-SNE plot, a random subset of 1000 PBMCs was used from each sample obtained by the mass cytometry analysis. NK natural killer, t-SNE t-distributed stochastic neighbor embedding, PBMC peripheral blood mononuclear cells, coef coefficient, *p p* value, fdr false discovery rate.

## TIGIT downregulated in CD57⁺CD4⁺ T cells in ≥2 Aab first group children

In the CD57⁺ subset of CD4⁺ T cells, TIGIT was lower in the ≥2 Aab first group compared to the controls (FDR = 0.006, Fig. 4a, c, Supplementary Data 2). This subset constitutes up to 0.52% of the total CD4⁺ T cell population and may represent cytotoxic T cells with CD45RO⁺PD-1⁺ profile. TIGIT, an inhibitory receptor expressed on lymphocytes, has been shown to negatively regulate T cell activation and suppress T cell functions through several mechanisms to maintain tolerance[19,20]. Thus, the lower level of TIGIT expression in CD57⁺CD4⁺ T cells of children developing type 1 diabetes may contribute to the disease progression. We did not find the differences in the TIGIT expression in either IAA- or GADA-first groups, and across all samples (Fig. 4c, Supplementary Data 2).

## CD39 elevated in CD4⁺ and CD8⁺ T cell subsets in IAA-first children

In CD4⁺ T cells, CD39 levels were higher in IAA-first group than in the controls. Such an increase was observed in two CD4⁺ T cell subsets (Supplementary Data 2), including CD25⁺CD127⁻ with the memory Treg phenotype (FDR = 0.007, Fig. 5a). CD39 expression was also elevated in HLA-DR⁺ICOS⁺ T cells (FDR = 0.004, Fig. 5b). In addition to the high

expression of HLA-DR and ICOS, the cells of this subset had a CD45RO⁺CCR7⁻PD-1⁺CD15s⁺CD38⁺ phenotype, possibly representing effector memory T cells. We did not observe differences in CD39 levels in CD4⁺ T cell subsets of ≥2 Aab or GADA-first groups, and across all samples (Fig. 5a, b, Supplementary Data 2).

We found that CD39 expression was also markedly increased in CD8⁺CD38⁺HLA-DR⁺ T cells in IAA-first children (FDR = 0.036, Fig. 5c, d, Supplementary Data 2). This subset constitutes about 1.97% of the total CD8⁺ T cell pool and has a CD45RO⁺CD28⁺ phenotype. We did not find the differences in the CD39 expression in either ≥2 Aab or GADA-first groups, and across all samples (Fig. 5d, Supplementary Data 2).

## CD161 increase in NK cells validated in an independent cohort

Using an independent cohort of children, we carried out validation experiments by flow cytometry. Due to sample material and technology limitations, we selected only two findings for validation based on their expression level and statistical significance: CD161 in NK cells and CD39 in Treg cells. The CyTOF results were validated by flow cytometry, confirming that CD161 expression in NK cells was higher in children positive for ≥2 Aab at initial seroconversion (n = 40, Fig. 6a, b) than in controls. Interestingly, the difference was more prominent and

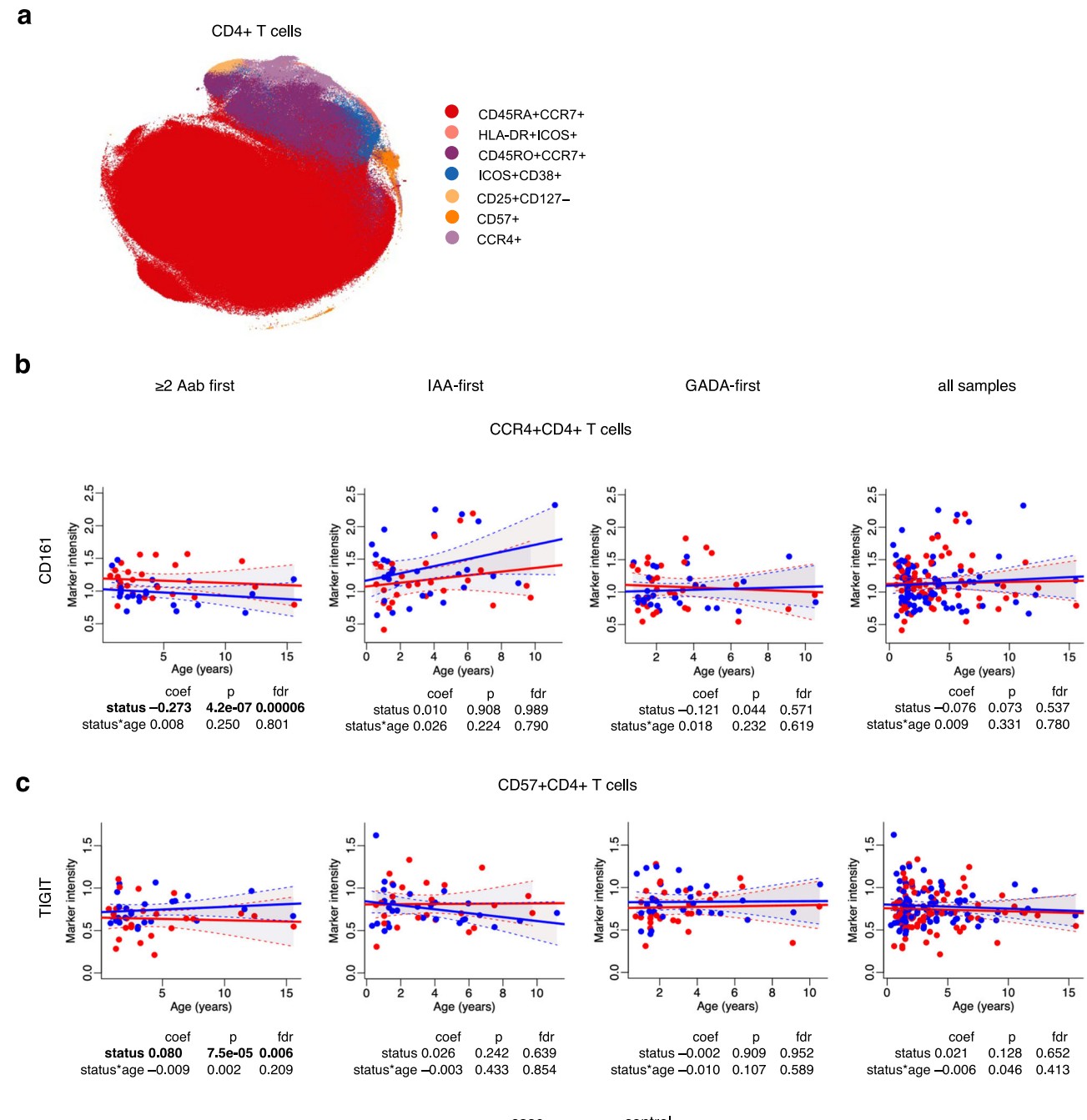

**Fig. 4 | CD161 upregulation and TIGIT downregulation in CD4⁺ T cell subsets in cases and controls of the ≥2 Aab first group. a** t-SNE plot of a random subset of 3500 CD4⁺ T cells from all samples obtained by the mass cytometry analysis and colored by the identified cell subtypes. **b** Scatter plot of CD161 expression levels (y-axis) in CCR4⁺CD4⁺ T cells over time (x-axis) in the children of the ≥2 Aab, IAA-, GADA-first and all samples groups. **c** Scatter plot of TIGIT expression levels (y-axis) in CD57⁺ CD4⁺ T cells over time (x-axis) in the children of the ≥2 Aab, IAA-, GADA-first and all samples groups. Each plot is annotated with the mixed effects model coefficient, *p*-value and FDR for the comparison between cases and controls (*status*) and their interaction with age (*status*age*). Red and blue dots indicate case and control samples, respectively. Solid red and blue lines show linear regression fit and shaded areas show 95% confidence intervals for the predicted values for cases and controls, respectively. Statistical analyses were performed using linear mixed effects modeling. Reported *p* values were calculated from two-tailed t-tests and multiple testing correction was done using the Benjamini–Hochberg method. t-SNE t-distributed stochastic neighbor embedding, coef coefficient, *p* *p* value, *fdr* false discovery rate.

significant at the very early stage of disease development, before seroconversion (Fig. 6b), confirming the results of the discovery cohort. However, we were not able to validate the results of CD39 expression (n = 28, Fig. 6c, d), perhaps due to the lower number of samples available for the analysis from the IAA-first individuals and their matched controls.

## Discussion

In this study, we found differences in the proportions of immune cell types between the cases and controls in each of three endotypes based on the autoantibodies that appear first. Interestingly, the analysis of the entire discovery cohort showed no differences in immune cell proportions in children progressing to type 1 diabetes and their

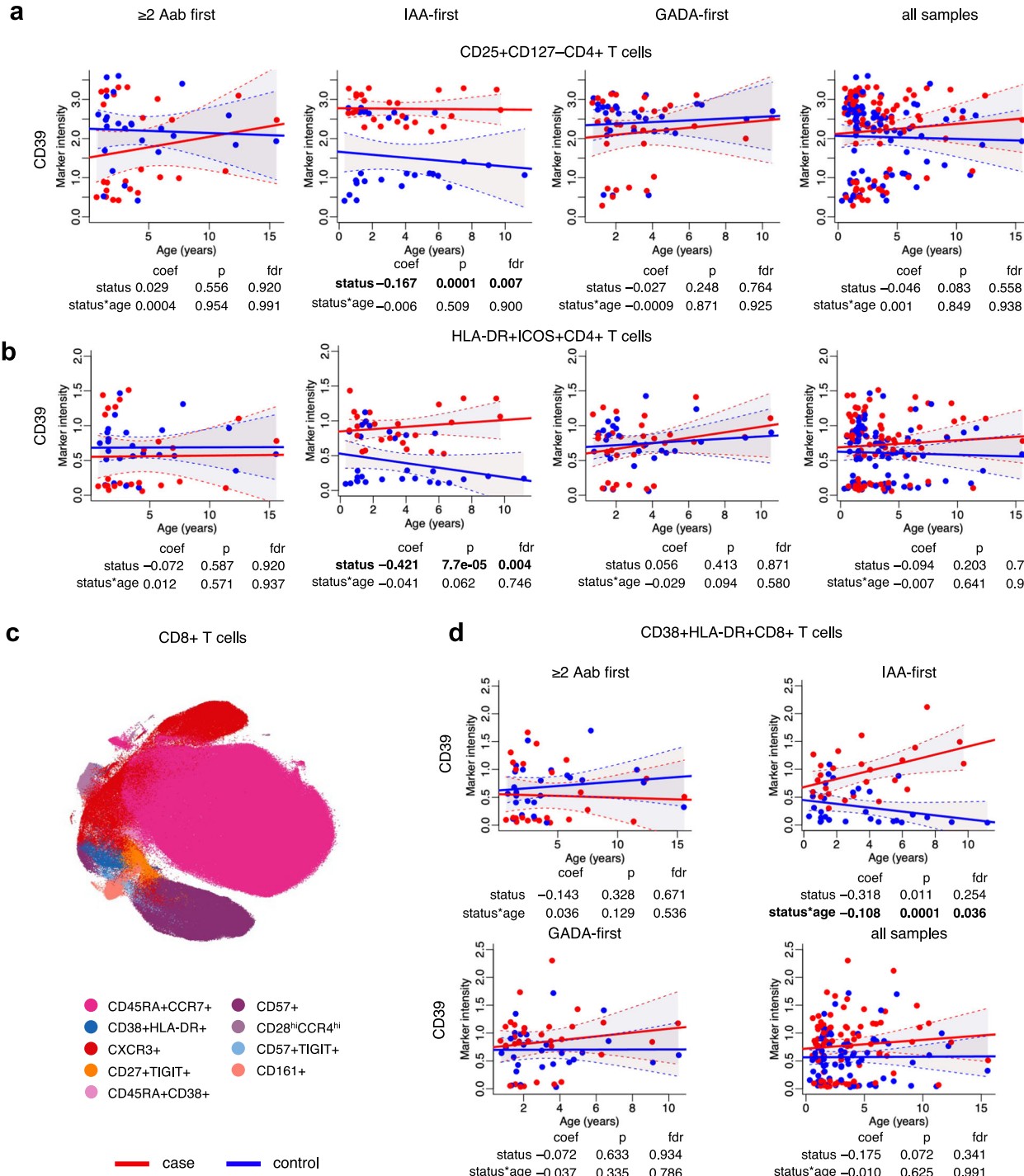

**Fig. 5 | Increased expression of CD39 in CD4⁺ and CD8⁺ T cells of IAA-first group.**
**a** Scatter plot of CD39 expression levels (y-axis) in CD25⁺CD127⁻ CD4⁺ T cells over time (x-axis) in the children of the ≥2 Aab, IAA-, GADA-first and all samples groups. **b** Scatter plot of CD39 expression levels (y-axis) in HLA-DR⁺ICOS⁺ CD4⁺ T cells over time (x-axis) in the children of the ≥2 Aab, IAA-, GADA-first and all samples groups. **c** t-SNE plot of a random subset of 3,500 CD8⁺ T cells from all samples obtained by the mass cytometry analysis and colored for the identified cell subtypes. The subsets and their phenotypes are indicated next to the t-SNE plots. **d** Scatter plot of CD39 expression levels (y-axis) in CD38⁺HLA-DR⁺ CD8⁺ T cells over time (x-axis) in the children of the ≥2 Aab, IAA-, GADA-first and all samples groups. Each plot is

annotated with the mixed effects model coefficient, *p*-value and FDR for the comparison between cases and controls (*status*) and their interaction with age (*status*age*). Red and blue dots indicate case and control samples, respectively. Solid red and blue lines show linear regression fit and shaded areas show 95% confidence intervals for the predicted values for cases and controls, respectively. Statistical analyses were performed using linear mixed effects modeling. Reported *p* values were calculated from two-tailed t-tests and multiple testing correction was done using the Benjamini–Hochberg method. coef coefficient, *p* *p* value, fdr false discovery rate.

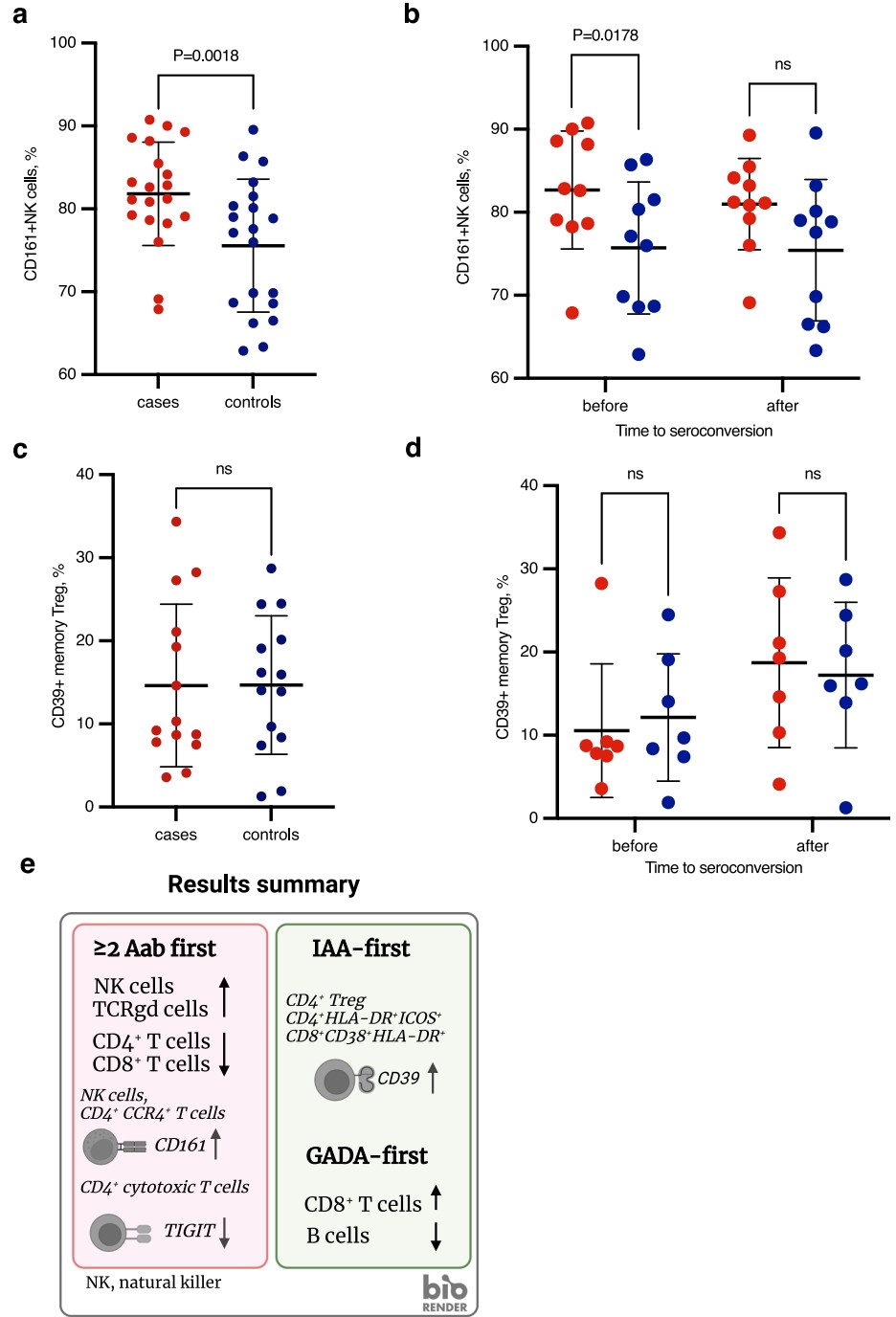

**Fig. 6 | Summary and validation of the results by flow cytometry. a** Strip chart of CD161-expressing NK cell proportions in cases (n = 20 longitudinally collected samples from 10 individual donors, red) and controls (n = 20 longitudinally collected samples from 10 individual donors, blue) of the ≥2 Aab first group in the validation cohort. *P* value = 0.0018. **b** Strip chart of CD161-expressing NK cell proportions in cases (red) and controls (blue) of the ≥2 Aab first group before (n = 10 case and 10 control individual donors) and after (n = 10 case and 10 control individual donors) seroconversion in the validation cohort. *P* value = 0.0178. **c** Strip chart of CD39-expressing Treg cell proportions in cases (n = 14 longitudinally collected controls.

samples from 7 individual donors, red) and controls (n = 14 longitudinally collected samples from 7 individual donors, blue) of the IAA-first group in the validation cohort. **d** Strip chart of CD39-expressing Treg cell proportions in cases (red) and controls (blue) of the IAA-first group before (n = 7 case and 7 control individual donors) and after (n = 7 case and 7 control individual donors) seroconversion in the validation cohort. Mean and standard deviation shown on the charts. Statistical analyses were performed using two-tailed paired t-test. ns not significant. **e** Schematic of the main study results. Created with BioRender.com. NK, natural killer.

This might be due to opposite trends in cell-type proportions in different autoantibody groups. These findings support the hypothesis of a heterogeneous pathogenesis of type 1 diabetes.

We also identified differentially expressed proteins in the immune cell subsets of children progressing to type 1 diabetes and control individuals at very early stages of disease development, even before

the appearance of type 1 diabetes-associated autoantibodies. Figure 6e summarizes the main results of the study. CD161 was upregulated more in the NK cells in children from the ≥2 Aab first group than in the NK cells from the controls. CD161 is a C-type lectin-like receptor in NK cells and its function remains to be elucidated, but IL-12 strongly upregulates CD161 in NK cells[21]. CD161 expression on human NK cells is

associated with a pro-inflammatory phenotype, and CD161[+] NK cells had higher IFNγ production upon IL-12 and IL-18 stimulation than their CD161[−] counterparts[22]. The increased frequency of NK cells with a pro-inflammatory phenotype in children progressing to type 1 diabetes may reflect and/or contribute to the ongoing disease process.

We also observed a higher expression of CD161 in CD4[+] T cells in cases than in controls among those with multiple autoantibodies. Interestingly, CD161 is reported to be a marker of human IL-17-producing CD4[+] T cells[23]. A study showed that circulating CD4[+] T cells from newly diagnosed type 1 diabetes patients secreted IL-17 in response to β cell autoantigens[24]. The elevated expression of IL-17E and IL-17F in circulating lymphocytes of type 1 diabetes patients, compared to controls, was also reflected in gene expression levels[25]. Further, CD4[+] T cells expressing CD161 have a transcriptional signature associated with Th1 and Th17 profiles[26]. Our results suggest that the IL-17 pathway is activated before seroconversion.

Next, we found a downregulation of TIGIT in cytotoxic CD57[+]CD4[+] T cells of children from the ≥ 2 Aab first group. The important role of TIGIT was recently highlighted in the context of autoimmune diseases[27]. The increased expression of immune response negative regulators was shown to limit β cell destruction by autoreactive T cells in a mouse model of type 1 diabetes[28]. Another study showed the accumulation of CD8[+] T cells expressing high levels of TIGIT and other co-inhibitory receptors in the individuals with best response to teplizumab treatment[29]. Thus, the decreased levels of TIGIT in cytotoxic CD57[+]CD4[+] T cells of children who later develop type 1 diabetes may play a role in the disease progression.

In children with IAA as the first appearing autoantibody, CD39 was increased in CD4[+] and CD8[+] T cell subsets albeit this was not confirmed in the validation cohort. The ectoenzyme CD39 is important for hydrolyzing ATP, which can trigger an inflammatory response, into immuno-suppressing adenosine. We found the most significant CD39 upregulation in memory Tregs and HLA-DR[+]ICOS[+] cells among other CD4[+] T cell subsets. In the Treg compartment, CD39 delineates a subset of cells characterized by potent immunesuppressive ability[30,31]. CD39 was linked to type 1 diabetes in a study, in which it was downregulated in memory Treg cells of adult patients (age 26.40 ± 6.90, mean ± SD) diagnosed with type 1 diabetes 1.5 to 17.8 years earlier[32]. However, the study design focusing on adults with already established type 1 diabetes was different from ours, which makes comparison to our results difficult. Further, a study on Crohn's disease reported the increased frequency of CD39[+]CD161[+]CD4[+] T cells in the peripheral blood and lamina propria of patients[33]. The increase of these cells was additionally associated with the clinical disease severity. Increased numbers of CD39[+]CD8[+] T cells were also observed in peripheral blood from patients with active Crohn's disease as compared to healthy control[34].

In conclusion, our results provide insights into immune cell composition and changes in expression of proteins associated with type 1 diabetes according to the type of autoantibody that appears first. Our study was limited by the number of individuals per each of the three autoantibody groups. However, this shortcoming was, to some extent, compensated for by the availability of samples from carefully matched controls and a sample series covering several time points during the time before seroconversion. If validated in a larger cohort of children, these findings may be useful for disease prediction and management. The current data supports the hypothesis that based on the type of autoantibody that appears first, there are distinct characteristics of early immune response in children who later develop clinical disease. Further, the results imply importance of considering endotypes in the analysis. In our study endotype-specific analysis revealed differences between cases and controls, which would have remained hidden, if all study subjects would have been combined.

## Methods

### Study cohort

All study subjects were participants in the DIPP study[16]. This ongoing, prospective follow-up study recruits newborns based on HLA-conferred genetic susceptibility to type 1 diabetes at Turku, Oulu and Tampere University Hospitals. The original study protocol screened the children for type 1 diabetes risk-associated HLA-DQB1*03:02 and DQB1*02 alleles along with protective DQB1*03:01 and DQB*06:02/3 alleles[35]. Since then, the screening has been improved by broadening the DQB1 genotyping to a "full-house" model and including selected DQA1 and DRB1 alleles, such as the DR4 subtypes[36]. Children carrying genetic predisposition to type 1 diabetes are invited to the DIPP follow-up study and, upon participation, monitored for the appearance of humoral β cell autoimmunity. In the early years of the study, children were first monitored only for the appearance of islet cell antibodies (ICA) and, after their appearance, also for IAA, GADA and antibodies against islet antigen-2 (IA-2A)[35]. Children born from 2003 have been monitored for all four autoantibodies at each study visit. During the first 2 years of life, study participants visit the clinic every 3–6 months and, after that, every 6–12 months. The follow-up continues until the age of 15. If type 1 diabetes-associated autoantibodies are detected at two consecutive clinic visits, the follow-up frequency is raised to every 3 months. Seroconversion is defined as positivity confirmed at two consecutive study visits. These seroconverted children are also monitored for their glucose metabolism. Diabetes is diagnosed according to the WHO and ADA criteria[37,38]. The study was originally approved by the Ethics Committee of the Hospital District of Southwest Finland, followed by the Ethics Committee of the Hospital District of Northern Ostrobothnia, and written informed consent was provided by the families participating in the study.

The discovery cohort comprised 29 case-control pairs, where the cases had all developed type 1 diabetes and the autoantibody-negative controls were matched for HLA, age and sex. 76% of total sample pairs were matched within 1 month, 8% within 2 months, 11% within 3 months, 3% within 5 months, and 1% within 4 and 6 months. Of note, for children aged 0 to 3 years, 89% of pairs were matched within 1 month. Eleven and nine of the selected cases had IAA or GADA as the first-appearing autoantibody, respectively. Nine children showed positivity for at least two autoantibodies at seroconversion. PBMC samples, collected at study clinic visits, were selected from time points of age of around 1 year, 3–6 months before seroconversion, 6–12 months after seroconversion, and approximately 1 year before the development of type 1 diabetes (Fig. 1A). The samples from autoantibody-negative controls were matched for time of sample acquisition with the closest available dates selected.

A validation cohort of 30 case-control pairs from the DIPP study was used to confirm the key findings from the discovery cohort. Seventeen of the cases had progressed to type 1 diabetes, and 13 were positive for multiple type 1 diabetes-associated autoantibodies. Two samples were analyzed for each case: one sample 3–6 months before and another 6–12 months after seroconversion. The IAA-first, GADA-first and ≥2 Aab first subcategories had 10 pairs each. In the IAA-first and GADA-first subgroups, the highly specific selection criteria for samples resulted in a shortage of suitable cases. Therefore, instead of a single case with two samples, "case pairs" were created with two very similar case children, for IAA-first (four case pairs) and GADA-first (one case pair) groups. The autoantibody-negative controls were selected using the same criteria as in the discovery cohort.

### Sample collection

PBMC samples were extracted from lithium heparin blood, collected at study clinic visits, using Ficoll-Paque Plus density gradient centrifugation. The samples were stored at −150 °C in RPMI 1640 medium with 10% human AB serum and 10% DMSO.

## Antibody labeling

The antibody conjugation was performed with Maxpar antibody labeling kit (Standard BioTools, USA), according to the manufacturer's protocol. To determine specificity and optimal concentration, we performed validation and titration for all conjugated antibodies. Further, to ensure minimal technical variation, we prepared a master mix of all conjugated antibodies, aliquoted and stored at −80 °C.

## Sample preparation and staining

Cryopreserved PBMC samples were thawed in a 37 °C water bath and washed with pre-warmed RPMI 1640 medium supplemented with 10% FCS and DNase-containing solution (CTL Anti-Aggregate Wash™ 20× Solution, Immunospot, USA). From each sample, $3 \times 10^6$ PBMCs were taken for the staining and washed with Maxpar Cell Staining Buffer (CSB) (Standard BioTools, USA). Next, FcR-blocking was done with Human TruStain FcX Fc receptor blocking solution (BioLegend, USA). We stained the samples with conjugated antibodies for 30 min at room temperature (RT). Cells were washed twice with CSB. The staining with Maxpar® Direct™ Immune Profiling Assay (MDIPA) kit (Standard Bio-Tools, USA) was performed as recommended in manufacturer's guidance. The cells were washed twice and fixed with freshly prepared 1.6% formaldehyde solution for 10 min at RT. Finally, the samples were incubated with Cell-ID intercalator solution (Standard BioTools, USA) at concentration of 125 nM overnight at 4 °C. Before acquisition, cells were washed twice with CSB and twice with Cell Acquisition Solution (Standard BioTools, USA). All antibodies are listed in Supplementary Table 1.

## Data acquisition

For the acquisition, cells were diluted to $0.5 \times 10^6$ cells/mL with Cell Acquisition Solution (Standard BioTools, USA) containing 10% EQ Four Element Calibration Beads (Standard BioTools, USA). Samples were analyzed on a Helios mass cytometer, (Standard Biotools, USA). For all samples, 300–600,000 events were collected (CyTOF Software v. 6). We aimed at collecting 600,000 events (cells) when possible, at least 300,000 events (cells) were collected from each sample as recommended by the Maxpar direct immune profiling assay manufacturer. This recommendation allows estimating low-abundance cell populations, such as pDCs, comprising 0.1–0.5% of total PBMC sample. For most of the samples 0.36, 0.37, 0.40 millions of live intact cells were analyzed for ≥2 AAb, IAA, and GADA first groups, respectively. The minimal reliable cell number per population was 100 intact live cells.

To minimize technical variations and improve data consistency, we took several measures. First, all the samples were handled, stained and acquired in a similar fashion, at the same site and Helios mass cytometer and by the same personnel and operator. Next, we prepared a master mix of additional antibodies for all the study cohort samples, aliquoted and stored them at −80 °C to ensure consistent staining pattern, as described here[39]. Further, all samples from each case-control pair were processed at the same time (including thawing, staining, and acquisition), i.e. in the same batch, and the model accounted for the batches by having the pair as a random effect. Additionally, we used the internal control, a PBMC sample, which was obtained from a single donor. The internal control sample was processed, stained and acquired along with the study samples of each case–control pair. t-SNE plots for the batch control samples can be found in Supplementary Fig. 2.

## Data analysis

Data cleanup was carried out using automated Maxpar Pathsetter™ software v. 2.0 (Standard BioTools, USA). Next, data analysis was performed using R language v. 4.3.0. The data were pre-processed using the flowCore (v. 2.12.2) package[40] and transformed using arcsin with a cofactor of 5. The cells were clustered using the FlowSOM (v. 2.8.0) package, followed by metaclustering with the ConsensusClusterPlus (v. 1.64.0) package[41].

The heterogeneity of PBMC was assessed in two consecutive steps. We first identified main immune cell types of PBMC samples, and further looked into heterogeneity of CD4⁺ and CD8⁺ T cells. In each step, we applied the combination of unsupervised (overclustering) and supervised clustering (removing and merging clusters) and manual cluster annotation based on known biology.

To identify B cells, CD8⁺ T cells, TCR-gd cells, NKT and MAIT cells, NK cells, pDCs, CD4⁺ T cells, basophils, monocytes, and mDCs, the following markers were used in clustering: CD45, IL-3R/CD123, CD19, CD4, CD8a, CD11c, CD16, CD161, CD56, γδ TCR, CD294, CD14, CD3, CD20, CD66b, and HLA-DR. The unsupervised clustering resulted in 20 clusters, 4 of the clusters were removed due to representing either debris (no lineage markers expressed) or doublets (showing expression of 2 or more lineage-specific markers, such as simultaneous CD3 and CD20 expression). In addition, we removed CD66b⁺ Granulocytes cluster, as we were not sure if that could be low-density neutrophils or contamination with conventional neutrophils during sample preparation procedure. The remaining clusters were finally merged based on lineage marker expression, resulting in 10 main cell types. After identifying the cell types, cell type proportions and cell type specific mean marker intensities were determined in each sample.

To examine changes in cell type proportions between the case and control groups, we used linear mixed effects modeling implemented in the lmerTest (v. 3.1-3) package[42]. Age, sex, HLA-DR3/4 status, and case-control status were treated as fixed effects, and processing batch and case-control pair as random effects, using the following formula: Cell type proportion ~ Age + Sex + HLA + Age*CaseCtrl + (1|Batch) + (1|Pair). The significance of the sample subsets (IAA-first, GADA-first, ≥2 Aab first groups) was further assessed using the following formula: Cell type proportion ~ Age + Sex + HLA + Age*Group + (1|Batch) + (1|Pair). Likelihood-ratio test was used to compare the goodness of fit of the two models for each cell type.

For each sample subset (IAA-first, GADA-first, ≥2 Aab first groups), marker intensities were statistically tested for differences between the case and control groups using linear mixed effects modeling. Age, sex, HLA-DR3/4 status, and case-control status were treated as fixed effects and case-control pair as a random effect. For each sample subset, we used the following formula: Marker intensity ~ Age + Sex + HLA + Age*CaseCtrl + (1|Pair). For the all-samples analysis, we used the following formula: Marker intensity ~ Age + Sex + HLA + Age*CaseCtrl + (1|Batch) + (1|Pair).

Further, we assessed the heterogeneity of CD4⁺ and CD8⁺ T cells using functional markers (i.e. secondary sub-clustering of main clusters). We focused on assessing these cell types in detail, as the marker panel was designed specifically for in-depth analysis of T cells. The clustering process was performed for the identified CD4⁺ T and CD8⁺ T cells using the following markers: ICOS, CD15s, CD39, CCR6, CTLA-4, IL-3R/CD123, CD4, CD16, CD45RO, CD45RA, CD161, CCR4, CD25, CD27, CD57, CXCR3, CXCR5, TIGIT, CD28, CD38, CD69, LAG-3, CD294, CCR7, CD3, CD20, CD66b, HLA-DR, PD-1, and CD127. Cell subtype proportions and their mean marker intensities were determined similarly as above and subjected to statistical testing.

FDR was corrected using the Benjamini−Hochberg method against all marker levels in all cell type clusters that were tested. The correction was done separately for the cell type proportions and for the additional CD4⁺ and CD8⁺ T cell subsets. Findings were considered to be significant at FDR < 0.05 and the minimal sufficient cell number per population was 100 intact live cells.

## Cluster annotation for CD4⁺ and CD8⁺ T cells

To characterize phenotypic diversity of CD4⁺ and CD8⁺ T cell compartment, we visualized the relevant functional markers on the t-distributed stochastic neighbor embedding (t-SNE) map of CD4⁺ and CD8⁺ T cells. Using unsupervised clustering, we identified seven and nine distinct CD4⁺ and CD8⁺ T cell subsets, respectively. Based on the

expression of CD45RA/RO, CCR7, CD27 and CD127, we identified CD4[+] T cells with naive, central memory, activated, regulatory, and effector phenotypes.

## Flow cytometry

Upon thawing, PBMCs were washed with FACS buffer (PBS, 2%FBS, 0,1%NaN3 sodium azide). Further, for CD161 detection in NK cells, PBMCs were incubated with the following antibody cocktail for 30 min at +4 C in the dark (anti-human CD45 FITC (1:100, clone HI30, BioLegend, cat # 304006), anti-human CD19 PE (1:100, clone HIB19, eBioscience, cat # 12-0199-42), anti-human CD3 PerCP-Cy5.5 (1:100, clone UCHT1, BD Biosciences, cat # 560835), anti-human CD14 PE-Cy7 (1:100, clone 63D3, BioLegend, cat # 367112), anti-human CD56 BV421 (1:100, clone NCAM16.2, BD Biosciences, cat # 562751), anti-human CD161 APC (1:100, clone HP-3G10, BioLegend, cat # 339912). For CD39 detection in memory Treg cells, CD4[+] T cells were isolated from PBMCs using human CD4 T cells kit (Dynabeads, Thermo Fisher). Next, isolated CD4[+] T cells were incubated with the following antibody cocktail for 30 min at +4 C in the dark (Anti-human CD3 PerCP-Cy5.5 (1:100, clone UCHT1, BD Biosciences, cat # 560835), anti-human CD4 BV421 (1:100, clone RPA-T4, BD Biosciences, cat # 562424), anti-human CD25 BB515 (1:20, clone BC96, BD Biosciences, cat # 567318), anti-human CD127 PE-Cy7 (1:100, clone A019D5, BioLegend, cat # 351320), anti-human CD45RA BV786 (1:100, clone HI100, BioLegend, cat # 304140), anti-human CD45RO PE (1:100, clone UCHL1, BD Biosciences, cat # 555493), anti-human CD39 APC (1:100, clone A1, BioLegend, cat # 328210). For both applications, staining with Fixable Viability Dye eFluor 780 (cat. 65-0865-14, eBioscience, Thermo Fisher Scientific, USA) was done after surface staining with antibodies. Further, cells were washed 2 times with FACS buffer. Data were acquired on the BD LSRFortessa™ Cell Analyzer flow cytometer (BD Biosciences, USA) with BD FACSDiva™ software v. 9. Data were analyzed using FCS Express 7 Flow software v. 7.16.0046 (De Novo Software, USA), the gating strategies for both applications shown in Supplementary Fig. 3 (CD161 in NK cells) and 4 (CD39 in memory CD4+ T cells).

## Reporting summary

Further information on research design is available in the Nature Portfolio Reporting Summary linked to this article.

## Data availability

The raw epidemiological and generated data are protected and are not available due to data privacy laws. The processed data underlying each figure and table are provided in the Supplementary Information and Source Data files. Source data are provided with this paper.

## Code availability

The R scripts generated during this study are available at Github repository (https://github.com/elolab/T1D-CyTOF-analysis).

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

## Acknowledgements

We are grateful to the families for their participation in the DIPP study and the clinical personnel for excellent collaboration, work with the families and collection of the samples for the study. We thank the personnel in DIPP autoantibody laboratory at the Department of Pediatrics, University of Oulu for their skillful work in measuring islet autoantibodies from the serum/plasma samples obtained at every DIPP study visit. We thank professor Olli Simell for his participation to the DIPP study. We thank Marjo Hakkarainen and Sarita Heinonen (Turku Bioscience Centre, University of Turku, Finland), Anne Suominen and Terhi Laakso (Institute of Biomedicine, University of Turku, Finland) for their excellent technical help. We acknowledge the Turku Bioscience Centre's core facility, the Cell Imaging and Cytometry Core (CIC), supported by Biocenter Finland, for their assistance and the Finnish Centre for Scientific Computing (CSC) for its efficient servers and data analysis resources. The DIPP study was supported by the following grants: JDRF [grants 1-SRA-2016-342-M-R, 1-SRA-2019-732-M-B, 3-SRA-2020-955-S-B]; European Commission [grant BMH4-CT98-3314]; Novo Nordisk Foundation; Academy of Finland [Decision No 292538] and Centre of Excellence in Molecular Systems Immunology and Physiology Research 2012-2017 [Decision No 250114]; Special Research Funds for University Hospitals in Finland; Diabetes Research Foundation, Finland; European Foundation for the Study of Diabetes; Päivikki and Sakari Sohlberg Foundation; Pediatric Research Foundation; and Sigrid Juselius Foundation, Finland. R.L. received funding from the Academy of Finland (grants 292335, 294337, 319280, 31444, 319280, 329277, 331790), Business Finland and by grants from the JDRF, the Sigrid Jusélius Foundation (SJF), Jane and Aatos Erkko Foundation, Finnish Diabetes Foundation and the Finnish Cancer Foundation. I.S. was supported by Turku Doctoral Programme of Molecular Medicine (TuDMM) and Finnish Diabetes Research Foundation, InFLAMES Flagship Programme of the Academy of Finland, Diabetes Wellness Suomi, Finnish cultural foundation. Prof. Elo reports grants from the European Research Council ERC (677943), Academy of Finland (310561, 314443, 329278, 335434, 335611 and 341342), and Sigrid Jusélius Foundation during the conduct of the study. JL and JI received funding from the Sigrid Jusélius Foundation, the Finnish Medical Foundation, the Finnish Pediatric Research Foundation and the Hospital Districht of South-West Finland.

## Author contributions

I.S. performed mass cytometry data acquisition and analysis, participated in the data analysis and interpretation of the results, prepared the figures, and wrote the manuscript. M.V. provided the clinical information on the study children, performed samples processing and participated in the drafting of the manuscript. S.Pi. and T.S. performed data analysis and participated in the drafting of the manuscript and preparing figures. S.Pa. provided the clinical information on the study children, performed samples processing. U.K. and O.R. participated in the interpretation of the results and the revising of the manuscript. E.R. participated in the sample processing. H.H., M.K., R.V., J.I., J.T., and J.L. were responsible for the DIPP study. J.L., M.K. L.L.E and R.L. designed the study. J.I. provided the samples and was responsible for the DNA isolation and HLA screening of the study children. J.T. provided the clinical information on the study children. J.L. supervised M.V. and S.P., provided the clinical information on the study children and participated in the interpretation of the results. L.L.E supervised S.Pi. and T.S. for data analysis and participated in the interpretation of the results. R.L. and J.L. initiated and supervised the study, interpreted the results and revised the manuscript. I.S., S.Pi., T.S. verified the underlying data. All authors revised the manuscript and approved the final version. These authors jointly supervised this work: J.L., L. L. E., R. L.

## Competing interests

The authors declare no competing interests.

## Additional information

[1]Turku Bioscience Centre, University of Turku and Åbo Akademi University, Turku, Finland. [2]InFLAMES Research Flagship Center, University of Turku, Turku, Finland. [3]Turku Doctoral Programme of Molecular Medicine, University of Turku, Turku, Finland. [4]Immunogenetics Laboratory, Institute of Biomedicine, University of Turku, Turku, Finland. [5]Faculty of Medicine and Health Technology, Tampere University, and Fimlab Laboratories, Tampere, Finland. [6]Research Program for Clinical and Molecular Metabolism, Faculty of Medicine, University of Helsinki, Helsinki, Finland. [7]Center for Child Health Research, Tampere University Hospital, Tampere, Finland. [8]Department of Pediatrics, Research Unit of Clinical Medicine, Medical Research Centre, Oulu University Hospital and University of Oulu, Oulu, Finland. [9]Centre for Population Health Research, University of Turku and Turku University Hospital, Turku, Finland. [10]Research Centre for Integrative Physiology and Pharmacology, Institute of Biomedicine, University of Turku, Turku, Finland. [11]Department of Pediatrics, University of Turku and Turku University Hospital, Turku, Finland. [12]Clinical Microbiology, Turku University Hospital, Turku, Finland. [13]Institute of Biomedicine, University of Turku, Turku, Finland. [14]These authors contributed equally: Inna Starskaia, Milla Valta, Sami Pietilä, Tomi Suomi. ✉e-mail: johanna.lempainen@utu.fi; laur-a.elo@utu.fi; rilahes@utu.fi

