## [Peer Review File · Nature Communications]

Distinct cellular immune responses in children en route to type 1 diabetes with different first-appearing autoantibodiesREVIEWER COMMENTS

Reviewer #1 (expert in type-1 diabetes):

The authors address the question of whether changes in frequencies of peripheral blood leukocyte populations are associated with pre-diabetes in children who differ in display of first islet autoantibody (AAb) specificities (GAD, insulin or > 2 AAb). They used mass and flow cytometry analysis of peripheral blood samples from children in the DIPP study who later progressed to T1D diagnosis and controls who did not progress. The main study conclusion is that differences in PBMC frequencies are apparent prior to T1D onset among these three groups of A AAb+ children at risk for T1D.

The rationale for the study is clear and is of interest. However, as outlined below, the analyses presented in the manuscript are not adequate to support the claims and conclusions.

1) Analyses with mixed linear models are key for the conclusions presented in the manuscript. In these linear mixed models, the authors use participant age at time of blood sampling, rather than timepoint (measurement 1,2,3) where the outcome of interest is T1D diagnosis. However, age has been shown to be associated with T1D risk, and therefore should be added as a covariate in the model.

2) Similarly, since HLA-DR3/4 status has been shown to associate with both T1D risk and with AAb specificities, genotype it should be added as a covariate in the model to address this potential confounding effect. Table 1 should include participant HLA-DR3/4 status.

3) Change in proportions of cell subsets with age: It is difficult to conclude that the frequencies or absolute numbers of CD4+ T cells, B cells, or NK cells change with age (Fig S1) in the absence of appropriate statistical tests, for example, an ordinal regression.

4) To identify changes in cell frequencies specific to an AAb subgroup is better handled by fitting two models on the cohort: i) one including an interaction term with the autoantibody subgroup, ii) one without an interaction term, and subsequently test which model better fits the data.

5) There is no information on the # of cells collected from each sample and most findings are based on proportion of cells. What was the minimal number of events (cells) collected to define each cell subgroup and how variable were they between participants within each AAb group?

6) For charts showing cell frequencies determined by flow cytometry, it is important to facet based on timepoint in addition to case vs. control.

7) The claim that CD161 expression on NK cells differs in the IAA-first group compared to other groups is not supported by statistical analysis.

Reviewer #2 (expert in mass cytometry and high-dimensional data analyses):

Summary

The authors undertake a 40-marker mass cytometric analysis of immune cell frequencies in samples from children at high risk of T1D prior to autoAb seroconversion from the DIPP study. Discovery analysis (n=30 pairs with age/sex/HLA-matched controls) analysed according to pattern of first-appearing autoAb (IAA, GADA and both; n=11,10 and 9 respectively). Validation cohort of n=30 (17 T1D+, 13 progressing to multiple autoAb) with validation of differential frequency of CD161+ NK cells between cases/controls in independent cohort.

Comments:

Given the clear potential for age to confound many of the effects reported, it would help to have a clearer explanation of the 'case-pair' strategy that was used as a back-up when there were

insufficient numbers of matched samples.

mixed modelling: unclear if the case:control pair was used as a random effect or if the sample id for either was used (the former approach could require exact matching of age, for example). Case/Control pairs could have a permissive matching allowing different ages within a threshold). Sample id should ideally be included as a random effect rather than combining both individuals into a single factor.

unsupervised hierarchical clustering: should clarify the rationale for the number of subclusters identified (how was this determined?)

The analysis is rather limited in terms of the number of sub clusters identified and greater population substructure is likely to be found with focus (for example) on each of the major clusters independently. This could enrich the analysis by identifying greater heterogeneity within subsets. The point is highlighted by the obvious heterogeneity of subpopulations within the 'NK' cluster in Figure 3.

CytoF batch effect: very common to identify batch structure. Should clarify how this was tested for and how it was handled to avoid confounding effects in the analysis.

Limited reporting of the mixed models makes it hard to understand what is being tested. The reported model structure could allow for interaction effects between case/controls or differences in % or overall changes with age. Full model reporting (formulae, coefficients, tests) would allow this to be clear. Currently, it isn't obvious what test (or what component of the model) the reported FDR value refers to.

The threshold for significance isn't clear but is presumed to be FDR 0.1 given the claims made. This should be clarified.

A central claim is that differences between case:control are themselves different between the 'endotypes' described. It would greatly help to present the plots shown throughout (e.g. in Fig 2, 5) side by side for each endotype so that they may be compared visually. It would make a more compelling case if the authors could show the result for each subgroup to allow direct comparison of the effects.

CD161 expression on NK cells - it appears that the effect also has an opposing association with age (increasing in controls, reducing in cases). Unclear if this has been formally tested or if this is tested in other comparisons.

Correction for multiple testing: It is unclear what correction has been made for significance testing in many places (e.g. Fig3A). Note the FDR should be corrected against all protein expression levels tested (presumably all that were measured) to confirm rationale for focussing on individual markers (such as CD161 and CD57). This critique applies to further subsetting of cell populations (Fig 3 onwards) where extensive subsetting occurs (e.g. to CD38+ICOS+ subsets). The identification of those subsets (in terms of the presumably unsupervised methods and thresholds used) and the extent of FDR correction (across all comparisons for all populations?) is not made clear.

This also applies to the measurement of CD57 expression on specific populations. It would help the reader to understand the relative expression levels of all proteins measured on each population and whether CD57/CD161 are the only differentials observed or if they are 'cherry-picked' from many. This could be relatively easily achieved by showing expression level per group for each marker and each group along with the model coefficients and significance tests performed. Without it, it becomes hard to be confident that differences reported are truly 'endotype-specific' and don't show a similar trend in other groups that is marginally non-significant.

This is particularly pertinent as the significance reported in differential expression of some markers is surprising given the figures illustrating them: for example, Fig4E shows overlapping 95%CI for distributions, yet a strongly differential expression level. This may be because other variables are accounted for in the mixed model and consequently it would help to see the full model output (as per above).

It also cannot be claimed that CD161 is higher in the IAA first group (Fig3E, FDR = 0.44): "higher levels of CD161 in NK cells were also detected in the IAA-first group than in the controls", p10.

Only 1 of the expression differences appears to validate with 1 other non-validated test for CD39+ Treg cells shown. I assume that the others were tested but did not validate...? This is important negative information to include and may mean that multiple correction needs to be applied to the validation testing also.

It is quite a conceptual leap to suggest that the pathophysiological mechanism may be similar in >2Aab and IAA-first groups on the basis of similar differential expression of a single protein marker on one cell subset....."Thus, children in ≥ 2 Aab first and IAA-first groups may share pathophysiological pathways leading to type 1 diabetes that are different from those seen in the GADA-first group". Overall my biggest criticism is that the authors report a single validated differential expression level of a single protein. It is therefore critical to understand how robustly this has been identified amongst the many candidates chosen (as per above) but also difficult to take this difference as strong support for differential pathogenesis between endotypes as is claimed.

Minor:

Data acquisition section: "Samples were analyzed by analysed" requires correction

RESPONSE TO REVIEWERS' COMMENTS

Reviewer #1 (expert in type-1 diabetes):

The authors address the question of whether changes in frequencies of peripheral blood leukocyte populations are associated with pre-diabetes in children who differ in display of first islet autoantibody (AAb) specificities (GAD, insulin or > 2 AAb). They used mass and flow cytometry analysis of peripheral blood samples from children in the DIPP study who later progressed to T1D diagnosis and controls who did not progress. The main study conclusion is that differences in PBMC frequencies are apparent prior to T1D onset among these three groups of A AAb+ children at risk for T1D.

The rationale for the study is clear and is of interest. However, as outlined below, the analyses presented in the manuscript are not adequate to support the claims and conclusions.

1) Analyses with mixed linear models are key for the conclusions presented in the manuscript. In these linear mixed models, the authors use participant age at time of blood sampling, rather than timepoint (measurement 1,2,3) where the outcome of interest is T1D diagnosis. However, age has been shown to be associated with T1D risk, and therefore should be added as a covariate in the model.

Our response:

We thank the reviewer for this comment. The age of the participant was used as a covariate in the model. The timepoints indicated in Fig. 1a correspond to the sample collection visits. The study design schematics (Fig. 1A) aimed to represent that from each participant we analysed 1-2 samples collected before seroconversion and 1-2 samples collected after seroconversion but before clinical disease. However, the schematic might have been confusing and, therefore, we now replaced it with a new diagram (Figure 1a) showing sample collection visits with respect to age and seroconversion. The figure legend was modified accordingly (page 21, lines 560-564).

2) Similarly, since HLA-DR3/4 status has been shown to associate with both T1D risk and with AAb specificities, genotype it should be added as a covariate in the model to address this potential confounding effect. Table 1 should include participant HLA-DR3/4 status.

Our response:

We thank the reviewer for this comment. HLA-DR3/4 status of participants was added to Table 1 (page 20). We have now included the HLA status in our models (page 14, lines 363-367) and updated the results accordingly.

3) Change in proportions of cell subsets with age: It is difficult to conclude that the frequencies or absolute numbers of CD4+ T cells, B cells, or NK cells change with age (Fig S1) in the absence of appropriate statistical tests, for example, an ordinal regression.

Our response:

The same statistical approach that was used for the functional markers was also applied for the cell type proportions. We have now added the linear mixed effect

model p-values and FDRs to the revised Supplementary Figure 1 and the manuscript (page 5-6, lines 110-114).

4) To identify changes in cell frequencies specific to an AAb subgroup is better handled by fitting two models on the cohort: i) one including an interaction term with the autoantibody subgroup, ii) one without an interaction term, and subsequently test which model better fits the data.

Our response:

We appreciate the reviewer's insight. However, we deliberately chose to limit the complexity of our models by not adding interaction terms to the models. Instead, while we had the subgroup as a random effect in the overall model, we chose to investigate each subtype separately. The reason was twofold: first, our main objective was not direct comparisons between subgroups, and second, the potential confounding effect of processing batches on these subgroups, which we considered as a random effect, made direct statistical comparisons challenging.

5) There is no information on the # of cells collected from each sample and most findings are based on proportion of cells. What was the minimal number of events (cells) collected to define each cell subgroup and how variable were they between participants within each AAb group?

Our response:

We thank the reviewer for the comment. We have now added the information on the number of cells collected to the revised manuscript. Generally, we aimed at collecting 600.000 events (cells) when possible, at least 300.000 events (cells) were collected from each sample as recommended by the Maxpar direct immune profiling assay manufacturer. This recommendation allows estimating low-abundance cell populations, such as pDCs, comprising 0.1–0.5% of total PBMC sample. The figure shows the number of live intact cells after data cleanup from all samples in each Aab group which were used for the analysis. For most of the samples 0.36, 0.37, 0.40 mln of live intact cells were analysed for ≥ 2 AAb, IAA, and GADA first groups respectively. The minimal reliable cell number per population was 100 intact live cells. This information was added to the revised manuscript (page 13, lines 320-326).

Number of live intact cells

6) For charts showing cell frequencies determined by flow cytometry, it is important to facet

based on timepoint in addition to case vs. control.

Our response:

We thank the reviewer for the suggestion. We have now added the charts which show the cell frequencies determined by flow cytometry at different timepoints (for the validation cohort, we had two timepoints, before and after seroconversion, revised Fig. 6b for CD161+NK cells and 6d for CD39+ memory Treg). The revised manuscript was modified accordingly (page 8, lines 186-189).

7) The claim that CD161 expression on NK cells differs in the IAA-first group compared to other groups is not supported by statistical analysis.

Our response:

We thank the reviewer for this comment. This claim was removed from the revised manuscript (page 6-7, lines 143-145).

“The CD161 levels in the NK cells were not different between the cases and controls of the IAA- and GADA-first groups.”

Reviewer #2 (expert in mass cytometry and high-dimensional data analyses):

Summary

The authors undertake a 40-marker mass cytometric analysis of immune cell frequencies in samples from children at high risk of T1D prior to autoAb seroconversion from the DIPP study.

Discovery analysis (n=30 pairs with age/sex/HLA-matched controls) analysed according to pattern of first-appearing autoAb (IAA, GADA and both; n=11, 10 and 9 respectively).

Validation cohort of n=30 (17 T1D+, 13 progressing to multiple autoAb) with validation of differential frequency of CD161+ NK cells between cases/controls in independent cohort.

Comments:

1) Given the clear potential for age to confound many of the effects reported, it would help to have a clearer explanation of the 'case-pair' strategy that was used as a back-up when there were insufficient numbers of matched samples.

Our response:

We thank the reviewer for this comment. We have a case-control study design. The case-control pairs were carefully matched by age. For most of the longitudinal timepoints, the age was matched as close as $\pm 1-2$ months. The matching age between cases and controls can be seen from the revised Figure 1a.

2) mixed modelling: unclear if the case:control pair was used as a random effect or if the sample id for either was used (the former approach could require exact matching of age, for example. Case/Control pairs could have a permissive matching allowing different ages within a threshold). Sample id should ideally be included as a random effect rather than combining both individuals into a single factor.

Our response:

We thank the reviewer for this comment. We used the case-control pair as a random effect in our models to maintain simplicity, given that the pairs were already rigorously matched based on several important factors, including age, sex, and HLA risk. In particular, the age of the case-control participants was closely matched $\pm 1-2$ months for the pairs.

3) unsupervised hierarchical clustering: should clarify the rationale for the number of subclusters identified (how was this determined?)

The analysis is rather limited in terms of the number of sub clusters identified and greater population substructure is likely to be found with focus (for example) on each of the major clusters independently. This could enrich the analysis by identifying greater heterogeneity within subsets. The point is highlighted by the obvious heterogeneity of subpopulations within the 'NK' cluster in Figure 3.

Our response:

We thank the reviewer for this comment. The heterogeneity of PBMC was assessed in two consecutive steps. We first identified the main immune cell types of PBMC

samples, and then further looked into heterogeneity of CD4+ and CD8+ T cells. In each step, we applied the combination of unsupervised (overclustering) and supervised clustering (removing and merging clusters) and manual cluster annotation based on known biology.

The main immune cell types of PBMCs were identified using lineage-specific markers. The unsupervised clustering resulted in 20 clusters, 4 of which were removed due to representing either debris (no lineage markers expressed) or doublets (showing expression of 2 or more lineage-specific markers, such as simultaneous CD3 and CD20 expression). In addition, we removed CD66b+ Granulocytes cluster, as we were not sure if that could be low-density neutrophils or contamination with conventional neutrophils during sample preparation procedure. The remaining clusters were finally merged based on lineage marker expression, resulting in 10 main cell types.

Further, we assessed the heterogeneity of CD4+ and CD8+ T cells using functional markers (i.e. secondary sub-clustering of main clusters). We focused on assessing these cell types in detail, as the marker panel was designed specifically for in-depth analysis of T cells.

These details have now been added to the revised manuscript (page 13-14, lines 344-348; 352-357; 368-370).

We also noted that Figure 3 had a wrong labeling. It shows PBMC composition, not only NK cells. This mistake was now corrected (Fig. 3c).

4) CyTOF batch effect: very common to identify batch structure. Should clarify how this was tested for and how it was handled to avoid confounding effects in the analysis.

Our response:

We thank the reviewer for this comment. To minimize technical variations and improve data consistency, we have taken several measures. First, all the samples were handled, stained and acquired in a similar fashion, at the same site and Helios mass cytometer and by the same personnel and operator. Next, we prepared a master mix of additional antibodies for all the study cohort samples, aliquoted and stored them at -80°C to ensure consistent staining pattern, as described here (Schulz, 2019). Further, all samples from each case-control pair were processed at the same time (including thawing, staining, and acquisition), i.e. in the same batch, and the model accounted for the batches by having the pair as a random effect. Additionally, we used the internal control, a PBMC sample, which was obtained from a single donor. The internal control sample was processed, stained and acquired along with the study samples of each case-control pair. This explanation on how we controlled for potential batch effects was added to the revised manuscript (page 13, lines 327-337). We also added t-SNE plots for the batch control samples (Supplementary Figure 2).

5) Limited reporting of the mixed models makes it hard to understand what is being tested. The reported model structure could allow for interaction effects between case/controls or differences in % or overall changes with age. Full model reporting (formulae, coefficients,

tests) would allow this to be clear. Currently, it isn't obvious what test (or what component of the model) the reported FDR value refers to.

Our response:

We thank the reviewer for this comment. We have added supplementary tables, showing formulae, coefficients, and tests for each analysis.

For the analysis of each group, we used the following model:

*Cell type proportions/Marker expression ~ Age + Sex + Age*CaseCtrl + HLA + (1|Pair)*

For the analysis of all samples, we used the following model:

*Cell type proportions/Marker expression ~ Age + Sex + Age*CaseCtrl + HLA + (1|Batch) + (1|Pair)*

6) The threshold for significance isn't clear but is presumed to be FDR 0.1 given the claims made. This should be clarified.

Our response:

We thank the reviewer for this comment. The threshold for significant differences in the revised manuscripts is $FDR < 0.05$. We have modified the text accordingly (page 14, lines 378-380).

7) A central claim is that differences between case:control are themselves different between the 'endotypes' described. It would greatly help to present the plots shown throughout (e.g. in Fig 2, 5) side by side for each endotype so that they may be compared visually. It would make a more compelling case if the authors could show the result for each subgroup to allow direct comparison of the effects.

Our response:

We thank the reviewer for this very good idea. We have added plots side by side for each endotype. The text and figures were modified accordingly.

8) CD161 expression on NK cells - it appears that the effect also has an opposing association with age (increasing in controls, reducing in cases). Unclear if this has been formally tested or if this is tested in other comparisons.

Our response:

We thank the reviewer for this comment. Based on the statistical testing, there was no association of CD161 expression with age ($FDR=0.9$), and also the interaction effect between the case/control status and age was not significant ($FDR=0.49$).

9) Correction for multiple testing: It is unclear what correction has been made for significance testing in many places (e.g. Fig3A). Note the FDR should be corrected against all protein expression levels tested (presumably all that were measured) to confirm rationale for focussing on individual markers (such as CD161 and CD57). This critique applies to further subsetting of cell populations (Fig 3 onwards) where extensive subsetting occurs (e.g. to CD38+ICOS+ subsets). The identification of those subsets (in terms of the presumably unsupervised methods and thresholds used) and the extent of FDR correction (across all comparisons for all populations?) is not made clear.

This also applies to the measurement of CD57 expression on specific populations. It would help the reader to understand the relative expression levels of all proteins measured on each population and whether CD57/CD161 are the only differentials observed or if they are 'cherry-picked' from many. This could be relatively easily achieved by showing expression level per group for each marker and each group along with the model coefficients and significance tests performed. Without it, it becomes hard to be confident that differences reported are truly 'endotype-specific' and don't show a similar trend in other groups that is marginally non-significant.

This is particularly pertinent as the significance reported in differential expression of some markers is surprising given the figures illustrating them: for example, Fig4E shows overlapping 95%CI for distributions, yet a strongly differential expression level. This may be because other variables are accounted for in the mixed model and consequently it would help to see the full model output (as per above).

Our response:

We thank the reviewer for this comment. FDR was corrected using the Benjamini–Hochberg method against all marker levels in all cell type clusters that were tested. The correction was done separately for the cell type proportions and for the additional CD4⁺ and CD8⁺ T cell subsets. This has now been described in the revised manuscript (page 14, lines 376-380). We have now also added the supplementary materials for all analyses and highlighted the significant results.

10) It also cannot be claimed that CD161 is higher in the IAA first group (Fig3E, FDR = 0.44): "higher levels of CD161 in NK cells were also detected in the IAA-first group than in the controls", p10.

Our response:

We thank the reviewer for this comment. This claim was removed from the revised manuscript (page 6-7, lines 143-145).

"The CD161 levels in the NK cells were not different between the cases and controls of the IAA- and GADA-first groups."

11) Only 1 of the expression differences appears to validate with 1 other non-validated test for CD39+ Treg cells shown. I assume that the others were tested but did not validate...? This is important negative information to include and may mean that multiple correction needs to be applied to the validation testing also.

Our response:

We thank the reviewer for pointing this out. We were not able to test all the findings in the validation cohort due to the technical limitations: the flow cytometry instrumentation at our Centre does not allow multicolor panel, which would be needed for validating several findings simultaneously; and the amount of PBMCs collected from children did not allow us to apply several flow cytometry panels per sample. Therefore, we selected only two findings for validation based on their statistical significance and expression level: CD161 in NK cells and CD39 in memory

Treg cells. This is now clarified in the revised manuscript (page 8, lines 182-184).

12) It is quite a conceptual leap to suggest that the pathophysiological mechanism may be similar in >2Aab and IAA-first groups on the basis of similar differential expression of a single protein marker on one cell subset....."Thus, children in ≥ 2 Aab first and IAA-first groups may share pathophysiological pathways leading to type 1 diabetes that are different from those seen in the GADA-first group". Overall my biggest criticism is that the authors report a single validated differential expression level of a single protein. It is therefore critical to understand how robustly this has been identified amongst the many candidates chosen (as per above) but also difficult to take this difference as strong support for differential pathogenesis between endotypes as is claimed.

Our response:

We thank the reviewer for pointing this out. The claim about similar pathophysiological mechanisms may be similar in >2Aab and IAA-first groups was removed from the manuscript.

We also thank the reviewer for paying attention to this. We selected only two findings for validation based on their statistical significance and expression level: CD161 in NK cells and CD39 in memory Treg cells for the reasons indicated above in our response to the comment #11.

Minor:

13) Data acquisition section: "Samples were analyzed by analysed" requires correction

Our response:

We thank the reviewer for the comment. The text was modified (page 13, line 319):

"Samples were analyzed on a Helios mass cytometer (Standard Biotools, USA)."

REVIEWER COMMENTS

Reviewer #1 (expert in type-1 diabetes):

We appreciate the changes to analysis and data presentation made by the authors in this revised manuscript.

A remaining concern is this Reviewer's point #4:

Reviewer: To identify changes in cell frequencies specific to an AAb subgroup is better handled by fitting two models on the cohort: i) one including an interaction term with the autoantibody subgroup, ii) one without an interaction term, and subsequently test which model better fits the data.

Author response: We appreciate the reviewer's insight. However, we deliberately chose to limit the complexity of our models by not adding interaction terms to the models. Instead, while we had the subgroup as a random effect in the overall model, we chose to investigate each subtype separately. The reason was twofold: first, our main objective was not direct comparisons between subgroups, and second, the potential confounding effect of processing batches on these subgroups, which we considered as a random effect, made direct statistical comparisons challenging.

Reviewer response to rebuttal: If the main objective of the study was not direct comparisons between the sub-groups, one cannot then claim that the differences observed are "endotype-specific". Regarding the concern that batch effects would confound comparison between groups, adding both terms to the model (batch, AAb group) should correct for any batch effect.

Therefore, the recommendation is that the revised manuscript either:

1. Provides a direct comparison between the subgroups as previously requested, or
2. Removes claims to have identified distinct "endotypes" from the manuscript.

Figures 2-5 would all benefit from an additional column showing analyses for the pooled samples.

Reviewer #2 (expert in mass cytometry and high-dimensional data analyses):

The authors have responded to most of the queries raised, providing satisfactory answers.

The changes made improve the clarity of the study description and make interpretation of the findings more straightforward for the reader.

The mixed models are now explicitly outlined and the included terms can be seen and understood, which is helpful. The inclusion of 'sample pair' as a random effect is reasonable, although this does make the proximity of the matching important (the model effectively assumes the pair is matched).

The authors clarify the pairs are matched but should quantify this: in response to Q1 they state "for most of the longitudinal timepoints, the age was matched as close as +/- 1-2 months". This is not precise enough given the model used: what % is "most"? and "as close as 1-2months" implies that weaker matching may be used in a substantial %.

I note that the response to Q2 claims that matching is within 1-2 months, seemingly in contrast to the response to Q1. I would advocate a simple, precise statement of the form " x% of sample pairs were matched within 2 months" and provide the range of disparities.

The other amendments, to clarify how batch structure is handled and unsupervised clustering is undertaken, are helpful.

Labelling the red/blue summary model fits as case/control (Figs2-5) would help the reader (rather than having to rely on the legend to specify).

Reviewer #2 (Remarks on code availability):

Only available on request

RESPONSE TO REVIEWERS' COMMENTS

Reviewer #1 (expert in type-1 diabetes):

We appreciate the changes to analysis and data presentation made by the authors in this revised manuscript.

A remaining concern is this Reviewer's point #4:

Reviewer: To identify changes in cell frequencies specific to an AAb subgroup is better handled by fitting two models on the cohort: i) one including an interaction term with the autoantibody subgroup, ii) one without an interaction term, and subsequently test which model better fits the data.

Author response: We appreciate the reviewer's insight. However, we deliberately chose to limit the complexity of our models by not adding interaction terms to the models. Instead, while we had the subgroup as a random effect in the overall model, we chose to investigate each subtype separately. The reason was twofold: first, our main objective was not direct comparisons between subgroups, and second, the potential confounding effect of processing batches on these subgroups, which we considered as a random effect, made direct statistical comparisons challenging.

Reviewer response to rebuttal: If the main objective of the study was not direct comparisons between the sub-groups, one cannot then claim that the differences observed are "endotype-specific". Regarding the concern that batch effects would confound comparison between groups, adding both terms to the model (batch, AAb group) should correct for any batch effect.

(1) Therefore, the recommendation is that the revised manuscript either:

1. Provides a direct comparison between the subgroups as previously requested, or
2. Removes claims to have identified distinct "endotypes" from the manuscript.

Our response:

We thank the reviewer for the suggestion. We have performed the direct comparison between the subgroups as recommended. The results are described in the revised manuscript (page 6, lines 116-138; page 14, lines 369-376) and all data analysis results can be found in the Supplementary Tables 2-4.

(2) Figures 2-5 would all benefit from an additional column showing analyses for the pooled samples.

Our response:

We thank the reviewer for this suggestion. We have now added the plots for pooled samples' analyses to Figures 2-5.

Reviewer #2 (expert in mass cytometry and high-dimensional data analyses):

The authors have responded to most of the queries raised, providing satisfactory answers.

The changes made improve the clarity of the study description and make interpretation of the findings more straightforward for the reader.

(1) The mixed models are now explicitly outlined and the included terms can be seen and understood, which is helpful. The inclusion of 'sample pair' as a random effect is reasonable, although this does make the proximity of the matching important (the model effectively assumes the pair is matched).

The authors clarify the pairs are matched but should quantify this: in response to Q1 they state "for most of the longitudinal timepoints, the age was matched as close as +/- 1-2 months". This is not precise enough given the model used: what % is "most"? and "as close as 1-2months" implies that weaker matching may be used in a substantial %.

I note that the response to Q2 claims that matching is within 1-2 months, seemingly in contrast to the response to Q1. I would advocate a simple, precise statement of the form " x% of sample pairs were matched within 2 months" and provide the range of disparities.

Our response:

We thank the reviewer for this suggestion. We have added the precise statement to the revised manuscript (page 11, lines 281-284):

"76% of total sample pairs were matched within 1 month, 8% within 2 months, 11% within 3 months, 3% within 5 months, and 1% within 4 and 6 months. Of note, for children aged 0 to 3 years, 89% of pairs were matched within 1 month."

The other amendments, to clarify how batch structure is handled and unsupervised clustering is undertaken, are helpful.

(2) Labelling the red/blue summary model fits as case/control (Figs2-5) would help the reader (rather than having to rely on the legend to specify).

Our response:

We thank the reviewer for this suggestion. We have now added the labelling to Figures 2-5.

Reviewer #2 (Remarks on code availability):

(3) Only available on request

Our response:

We thank the reviewer for this comment. We have compiled R scripts (<https://github.com/elolab/T1D-CyTOF-analysis>) and added the link for public repository to the revised manuscript (page 15, lines 410-412).

REVIEWERS' COMMENTS

Reviewer #1 (Remarks to the Author):

The authors have performed the comparisons we requested between purported sub-groups and display the data clearly for readers to judge the significance of the observed differences between groups, where present. The data displays will allow direct comparison of results of this study to future studies in other cohorts which will be required to determine whether the proposed "endotypes" are robust and validated in other study cohorts.

Reviewer #2 (Remarks to the Author):

The authors have directly addressed the remaining query from Reviewer 1 and I would recommend publication.

RESPONSE TO REVIEWERS' COMMENTS

Reviewer #1 (Remarks to the Author):

The authors have performed the comparisons we requested between purported sub-groups and display the data clearly for readers to judge the significance of the observed differences between groups, where present. The data displays will allow direct comparison of results of this study to future studies in other cohorts which will be required to determine whether the proposed "endotypes" are robust and validated in other study cohorts.

Our response:

We thank the reviewer for this comment.

Reviewer #2 (Remarks to the Author):

The authors have directly addressed the remaining query from Reviewer 1 and I would recommend publication.

Our response:

We thank the reviewer for this comment.